# Improving Endogenous Nitric Oxide Enhances Cadmium Tolerance in Rice through Modulation of Cadmium Accumulation and Antioxidant Capacity

Wei Cai [1], Wenshu Wang [1], Hui Deng [1], Bin Chen [1], Guo Zhang [1], Ping Wang [1], Tingting Yuan [2,*] and Yongsheng Zhu [1,*]

1 Institute of Crop Science, Wuhan Academy of Agriculture Science, Wuhan 430000, China; weicai@whu.edu.cn (W.C.)
2 State Key Laboratory of Hybrid Rice, College of Life Sciences, Wuhan University, Wuhan 430072, China
* Correspondence: yuantingting@whu.edu.cn (T.Y.); yongshengzhu@163.com (Y.Z.); Tel.: +86-27-84874557 (Y.Z.)

**Abstract:** Nitric oxide (NO) plays an important role in plant stress responses. However, the mechanisms underlying NO-induced stress resistance to cadmium (Cd) stress in rice remain elusive. In this study, rat neuron NO synthase (nNOS)-overexpressing rice plants with higher endogenous NO level showed higher cadmium stress tolerance than the wild-type plants. The results showed that *nNOS*-overexpressing rice plants accumulated less cadmium in the roots and shoots by downregulating the expression of Cd uptake and transport related genes including *OsCAL1*, *OsIRT2*, *OsNramp5*, and *OsCd1*. Moreover, *nNOS*-overexpressing rice plants accumulated less hydrogen peroxide ($H_2O_2$), accompanying with higher expression of antioxidant enzyme genes (*OsCATA*, *OsCATB*, and *OsPOX1*) and corresponding higher enzyme activities under cadmium stress. Furthermore, the transcription of melatonin biosynthetic genes, including *OsASMT1*, *OsTDC1*, *OsTDC3*, and *OsSNAT2*, was also upregulated in *nNOS*-overexpressing plants, resulting in increased content of melatonin under cadmium treatment compared with the wild-type controls. Taken together, this study indicates that *nNOS* overexpression improves Cd tolerance of rice seedlings through decreasing cadmium accumulation and enhancing the antioxidant capacity and melatonin biosynthesis of the plants.

**Keywords:** nitric oxide; cadmium stress; reactive oxygen species; melatonin





## 1. Introduction

Rice is one of the most important crops in Asia. However, rice production safety is threatened by the toxic heavy metal cadmium (Cd), due to the increasing problem of Cd pollution. When rice plants are grown in Cd-polluted soil, they can absorb Cd through their roots, which is then transported to the shoots and grains. The excessive accumulation of Cd in soil not only inhibits rice growth, it also endangers human health through the food chain [1]. In order to resist Cd toxicity, rice has evolved many resistance strategies. On the one hand, rice reduces Cd accumulation by affecting Cd uptake, transport and chelation. In recent years, many Cd uptake- and transport-related genes, including *OsCAL1* (cadmium accumulation in leaf 1), *OsIRT2* (iron-regulated metal transporter2), *OsNramp5* (natural resistance-associated macrophage protein 5), and *OsCd1* (cadmium transporter gene 1), have been identified [2]. On the other hand, rice can reduce cell damage caused by Cd-induced overexpressed reactive oxygen species (ROS) by activating antioxidant enzymes such as superoxide dismutase (SOD), catalase (CAT) and peroxidase (POX) [1]. Thus, improving the cadmium tolerance and decreasing the cadmium accumulation in rice through biotechnology has become an urgent task given the increasing problems related to cadmium pollution.

It has been reported that many signaling molecules in plants are involved in the Cd stress response. Nitric oxide is a signaling molecule that is involved in plant growth and

development processes, as well as in responses to environmental stresses [3–5]. Recently, researchers have increasingly been reporting that NO modulates the resistance of plants to Cd stress by affecting physiological metabolic processes such as reactive oxygen species (ROS), photosynthesis, chlorophyll synthesis, and cadmium uptake [6–8], but the role of NO in response to Cd stress is still disputed. The Cd-stress-induced change in NO levels in many plants is influenced by Cd concentration, treatment time, and plant species [9–11]. For example, the NO level was substantially increased in the roots of barley and rice under Cd treatment [12,13], while Cd stress inhibited endogenous NO generation in peas and rice [14,15]. Additionally, NO may play different roles in the same biological process in different plants. For instance, inhibition of NO accumulation by means of NO scavenger 2-[4-carboxyphenyl]-4,4,5,5-tetramethylimidazoline-1-oxy-3-oxide (c-PTIO) or NOS inhibitor $N^{\omega}$-nitro-L-arginine-methylester(L-NAME) was shown to result in the prevention of Cd-stress-induced oxidant damage in Arabidopsis and yellow lupine [16,17], but application of the NO donor sodium nitroprusside (SNP) decreased ROS accumulation in Cd-stressed *Brassica juncca* and rice seedlings [18,19]. In addition to the above-mentioned NO-mediated Cd tolerance in plants, NO also serves as a gas messenger, and is involved in signaling transduction by regulating relevant gene expression and S-nitrosylation modifications of target proteins [20,21].

Moreover, NO can also regulate hormone homeostasis such as indole-3-acetic acid (IAA) or melatonin (N-acetyl-5-methoxytryptamine), so as to alleviate Cd toxicity in plants [22,23]. In recent decades, an increasing number of studies have reported that Cd stress can induce melatonin accumulation, and exogenous melatonin can improve Cd tolerance in different plants [24]. Melatonin is synthesized via four continual enzymatic reactions from tryptophan, requiring at least six enzymes: tryptophan decarboxylase (TDC), tryptophan hydroxylase (TPH), tryptamine 5-hydroxylase (T5H), N-acetylserotonin methyltransferase (ASMT), and serotonin N-acetyltransferase (SNAT) [24]. Many results have been reported showing that exogenous melatonin can alleviate Cd-induced oxidative damage by activating antioxidant systems [25,26], and can decrease Cd accumulation by regulating the transcription of iron-transport genes [25,27]. Although melatonin and NO play similar roles in response to Cd stress in plants, the relationship between them is still unclear.

Knowledge about the roles of NO in plants has largely been obtained through the exogenous application of NO donors such as SNP, the NO scavenger c-PTIO, and the NOS inhibitor L-NAME, which might be affected by differences in the concentration and time point of application of chemical treatments [3]. In addition, whether the application of NO donors or scavengers reflects the physiological status of NO is still unclear [28,29]. Therefore, plant materials with different endogenous NO contents should be used to assess the function of NO in response to Cd stress. Although many reports indicate that higher plants possess arginine-dependent NO synthase (NOS) activity, no NOS coding gene has yet been found in higher plants. Lin et al. (2012) reported a rice mutant *Osnoe1* that showed increased NO levels. *OsNOE1* encodes a rice catalase OsCATC, but whether its role arises from $H_2O_2$ or NO accumulation—or their crosstalk—remains elusive [30]. Therefore, it may be more accurate to investigating in the vivo roles of NO in plants through specific modulation of endogenous NO levels with no significant effect on plant development.

Overexpression of rat *nNOS* increases both NOS activity and NO content in transgenic Arabidopsis, tobacco, and rice plants, thus increasing tolerance to stresses from drought, salt and pathogens [31–33]. To assess the role of endogenous NO in response to Cd stress, *nNOS*-overexpressing rice plants are used in the current study. Our results indicate that the *nNOS* overexpression in rice plants improves Cd tolerance and decreases cadmium accumulation. Furthermore, the transgenic rice plants show enhanced antioxidant capacity, higher melatonin content, and changed expression of related genes under Cd stress.

## 2. Materials and Methods

### 2.1. Plant Materials and Growth Conditions

Rice (*Oryza sativa* L. cv. Zhonghua11) was used both as the wild type and for the generation of *nNOS* transgenic plants. Rice seeds were sterilized with 70% (*v/v*) ethanol for 5 min and subsequently with 5% (*w/v*) NaClO for 30 min, washed at least three times with sterile water, and then plated on agar medium containing 1/2 MS medium in plant growth chambers (50% humidity, 200 µmol m$^{-2}$s$^{-1}$, 14 h light/10 h dark cycle, and 28–30 °C).

### 2.2. Stress Treatments and Plant Sampling

To determine a suitable Cd treatment concentration, we transferred 2-day-old wild-type (WT) rice seedlings germinated on half-strength Murashige and Skoog (1/2 MS) plates to new plates containing 0, 50, 100, or 200 µM CdCl$_2$, and the seedlings were photographed and root and shoot lengths were measured using Image J software (Version 8.0) at 1, 2, 3, 4, and 5 days after transfer.

To test the effects of exogenous SNP treatment on the root and shoot growth under normal and cadmium-stress conditions, we transferred germinated rice seedlings to 1/2 MS medium containing 0, 20, 50, or 100 µM SNP with or without 200 µM CdCl$_2$, and the seedlings were photographed and root and shoot lengths measured at 1, 2, 3, 4, and 5 days after transfer.

To evaluate the plants' tolerance to cadmium stress, rice seeds were plated on agar medium containing 1/2 MS medium for 2 days in a plant growth chamber. Uniformly germinated rice seedlings were then transferred to 1/2 MS medium supplemented with 200 µM CdCl$_2$. After 1, 2, 3, 4, and 5 d of growth, the seedlings were photographed and root and shoot lengths were measured. At least 24 seedlings were analyzed per treatment.

To measure the transcript levels of selected genes and physiological parameters including H$_2$O$_2$ content, CAT and POX activity, chlorophyll content, cadmium content, and melatonin content under Cd stress, the roots, shoots, or whole seedlings of the tested plants under the different treatment conditions were sampled at the designated time for further analysis.

### 2.3. Measurement of NO Content

One-week-old wild-type rice seedlings were treated in 200 µM CdCl$_2$, and the NO content in the roots was assayed using the specific fluorescent probe DAF-FM DA at 24 h post treatment [33]. For DAF-FM DA imaging, the primary roots of the seedlings were incubated in 2 mL EP tubes with 10 µM DAF-FM DA in 20 mM HEPES-NaOH, pH 7.5, for 1 h, and rinsed three times with sterile water. Then, the samples were examined under an Olympus BX60 (Olympus, Tokyo, Japan) differential interference contrast microscope equipped with a CCD Olympus dp72 camera (Olympus, Tokyo, Japan) with an excitation of 488 nm and an emission of 515 nm. At least 24 seedlings per treatment were analyzed.

### 2.4. Measurement of NOS Activity

NOS activity was measured as previously described [33]. Briefly, approximately 0.5 g of rice seedlings was frozen and ground with liquid nitrogen, and then extracted with 2 mL buffer (50 mM Tris-HCl, pH 7.4, 1 mM EDTA, 1 mM dithiothreitol, 1 mM leupeptin, 1 mM pepstatin, and 1 mM phenylmethylsulfonyl fluoride). After centrifuging at 12,000× *g* for 15 min at 4 °C, the supernatant was used as the enzyme extract. NOS activity was assayed using the NOS assay kit (Beyotime, Shanghai, China), following the manufacturer's instructions.

### 2.5. Measurement of H$_2$O$_2$ Content, CAT Activity and POX Activity

The rice seedlings treated with or without 200 µM CdCl$_2$ for 5 days were used for the measurement of H$_2$O$_2$ content, CAT activity and POX activity. H$_2$O$_2$ content was measured using the Hydrogen Peroxide Assay Kit (Beyotime, Shanghai, China), following the manufacturer's instructions.

To measure the activities of CAT and POX, the total protein from seedlings was extracted using 0.05 M potassium phosphate buffer (pH 7.0), and the extract was centrifuged at $12{,}000 \times g$ for 15 min at 4 °C. Subsequently, the supernatant was used as the enzyme extract. CAT activity was detected using a Catalase Assay Kit (Beyotime), in accordance with the manufacturer's instructions. POX activity was measured as described previously [33]. The reaction mixture contained 0.1 mL of enzyme extract, 0.029 M potassium phosphate buffer (pH 5.5), 0.1% ($v/v$) $H_2O_2$ and 0.01 M guaiacol as substrates. The oxidation of guaiacol was monitored by the absorbance measured at 470 nm every 10 s.

### 2.6. Quantitative Real-Time PCR

Rice seedlings treated with or without 200 μM $CdCl_2$ for 5 days were sampled for qRT-PCR as previously described [33]. The total RNA was extracted from rice leaves using TRIzol reagent (Invitrogen, CA, USA). We used 1 μg of total RNA treated with RQ1 RNase-free DNase (Promega, Beijing, China) for cDNA synthesis with an RT kit (Toyobo, Shanghai, China) in accordance with the manufacturer's instructions. Quantitative real-time PCR assays were performed on a Bio-Rad CFX96 apparatus with the dye SYBR Green I (Invitrogen). The rice gene *eEF1α* was chosen as the internal control for the following analysis. The gene-specific primers are listed in the Supplemental Data, Table S1.

### 2.7. Measurement of Cd Content

The Cd content was analyzed according to a method described previously [34]. Briefly, rice seedlings treated with or without 200 μM $CdCl_2$ for 5 days and roots and shoots were sampled; then, the samples were washed with sterile water and then dried at 80 °C for 1 d. The dried samples were digested with $HNO_3/HClO_4$ (4:1, $v/v$) at 180 °C and then the digested solution was diluted with sterile water for determination of Cd content using an atomic absorption spectrometer.

### 2.8. Measurement of Chlorophyll Content

The chlorophyll content was determined according to a previously described method [35]. Briefly, 0.5 g fresh leaves of plants from different lines which treated with or without 200 μM $CdCl_2$ for 5 days were collected and incubated in 20 mL of 80% ($v/v$) acetone and kept in darkness for 24 h. After centrifugation, the extracted solutions were used for the total chlorophyll content determination.

### 2.9. Measurement of Melatonin Content

The melatonin content in rice tissues was determined as described previously [36]. Briefly, after treatment with or without 200 μM $CdCl_2$ for 5 days, root and shoot samples were extracted with acetone: methanol: water ($v{:}v{:}v$ = 89:10:1) and centrifuged. Subsequently, the supernatant was used for the melatonin content determination using a melatonin enzyme-linked immunosorbent assay (ELISA) kit.

### 2.10. Statistical Analysis

All experiments were performed in at least three independent biological replicates and three technical repetitions. The significance of differences was determined with ANOVA or Student's *t*-test, as indicated in the figure legends.

## 3. Results

### 3.1. Exogenous NO Donor Alleviated Cd Toxicity in Rice Seedlings

To elucidate how NO mediates cadmium stress in rice, we examined the root and shoot growth in different Cd concentrations. As shown in Supplementary Figure S1A,B, the root and shoot length were markedly inhibited by 200 μM $CdCl_2$ compared with the untreated control. Therefore, 200 μM $CdCl_2$ was used in the subsequent experiments based on the above results.

Nitric oxide may regulate plant growth in a dose-dependent manner. We tested this hypothesis through experiments employing the exogenous application of different concentrations of NO donor SNP. As shown in Figure 1A,B, the root and shoot lengths were markedly reduced in the 50 and 100 μM SNP treatment groups compared with the untreated controls, whereas the plants treated with 20 μM SNP showed no significant changes compared with those in the control group. Moreover, the application of Cd in the 50 and 100 μM SNP treatment groups abrogated this effect, whereas Cd and 20 μM SNP alleviated the inhibitory effects on the growth of the roots and shoots compared with Cd treatment alone (Figure 1C,D). Therefore, 20 μM SNP was used in subsequent experiments.

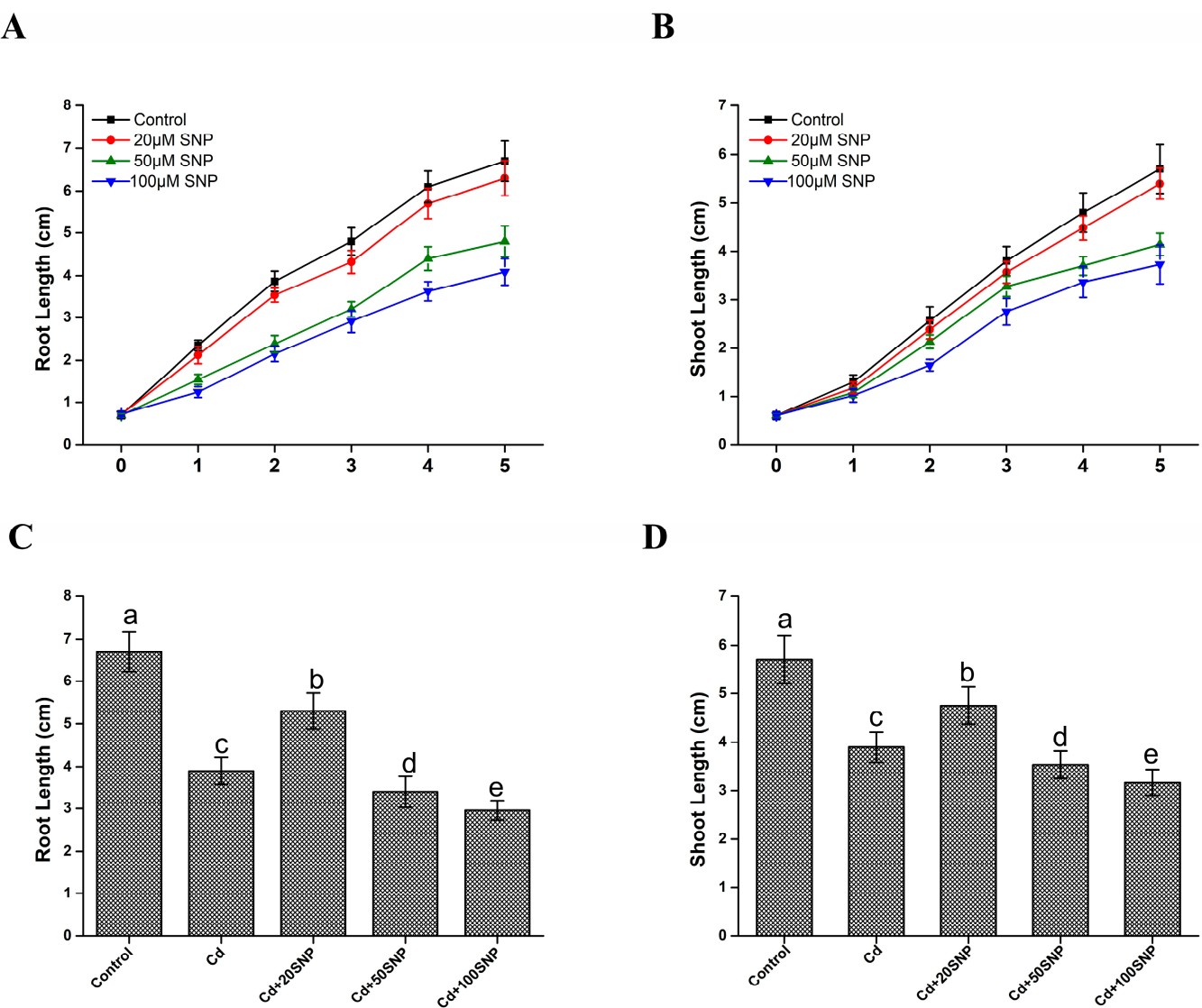

**Figure 1.** Effects of exogenous SNP treatment on root and shoot growth under normal and cadmium-stress conditions. Two-day-old seedlings germinated on 1/2MS plates were transferred to new plates new plates containing 0, 20, 50, or 100 μM SNP, and then root (**A**) and shoot (**B**) lengths were measured at 1, 2, 3, 4, and 5 days after transfer. (**C,D**) The root (**C**) and shoot lengths (**D**) of rice seedlings were measured after 5 days of growth on the medium (1/2 MS containing 0, 20, 50, or 100 μM SNP with 200 μM CdCl$_2$).The results shown are the means ± SD. Values were derived from three independent biological experiments. Different letters indicate significantly different values ($p < 0.05$ by Tukey's test). Cd, 200 μM CdCl$_2$; 20 SNP, 20 μM NP; 50 SNP, 50 μM SNP; 100 SNP, 100 μM SNP.

To verify whether Cd changed the NO level in rice seedlings, 1-week-old wild-type rice seedlings were treated in 200 μM CdCl$_2$, and the NO level in the roots was assayed at 24 h post treatment. The results of fluorescence analysis showed increased NO levels induced by cadmium, which were strongly reduced by an NOS inhibitor (L-NAME) (Figure 2A), suggesting that Cd stress can modulate NO content, possibly through NOS activity.

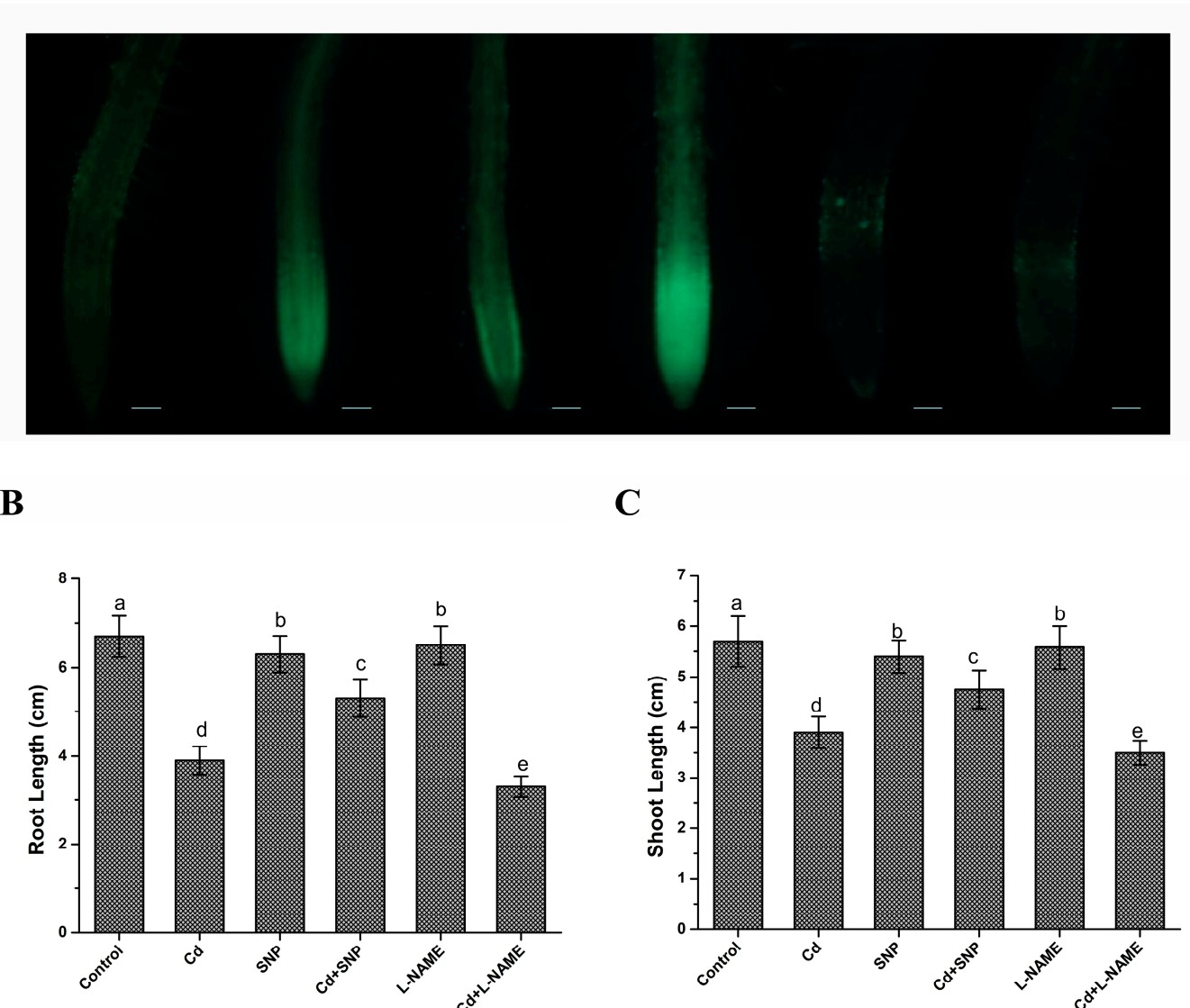

**Figure 2.** Endogenous NO level in response to Cd stress and the effect of exogenous NO donor on Cd toxicity in rice seedlings. (**A**) Effect of cadmium stress on the endogenous NO level in the roots of wild-type rice plants. One-week-old wild-type plants were treated with 200 μM CdCl$_2$ combined with or without 1 mM L-NAME for 24 h, and then NO production was measured. NO fluorescence in roots of the *nNOS*-overexpressing lines and wild-type plants examined using DAF-FM DA. (**B**,**C**) Effects of SNP or L-NAME on the root (**B**) and shoot (**C**) growth under cadmium stress. The results shown are the mean ± SD. Values were derived from three independent biological experiments. Different letters indicate significant differences using Tukey's multiple comparison test at *p* < 0.05. Bars = 50 μm; Cd, 200 μM CdCl$_2$; SNP, 20 μM SNP; L-NAME, 1 mM L-NAME.

Then, the effects of exogenous NO treatment on Cd resistance in rice seedlings were examined. Germinated wild-type rice seedlings were transferred to 1/2 MS medium containing 200 μM CdCl$_2$ with or without 1 mM L-NAME or 20 μM SNP, and then root and shoot lengths were measured 5 d after transfer. As shown in Figure 2B,C, the combined application of CdCl$_2$ and SNP reduced the inhibitory effect of cadmium treatment on the root and shoot lengths, whereas CdCl$_2$ and L-NAME treatment abrogated the inhibitory effect on the growth of roots and shoots compared with cadmium treatment alone. Taken together, NO is induced by Cd stress and exogenous NO donor can alleviate Cd toxicity in rice seedlings. These findings suggest that NO plays a positive role in the cadmium-mediated inhibition of the growth of rice seedlings.

### 3.2. nNOS-Overexpressing Rice Plants Demonstrated Improved Tolerance to Cadmium Stress

Due to the lack of materials with altered NO levels, the role of endogenous NO in rice responses to Cd stress is still limited. As such, in this study, the tolerance of *nNOS*-overexpressing rice plants 35S:*nNOS* (#2, #8 and #20), which had higher NOS activity and NO content, to Cd stress was investigated (Figure 3A,B). For this purpose, 2-day-old seedlings of both wild-type and transgenic lines were transferred onto 1/2 MS medium with or without supplementation with 200 μM CdCl$_2$, and both shoot and root lengths were assayed at 5 days after transfer. Although all the tested transgenic lines exhibited shoot and root lengths similar to those of the wild-type lines when grown in normal 1/2 MS medium, the transgenic lines were less sensitive to cadmium stress in terms of changes in shoot and root lengths (Figure 3C–F). Under Cd stress, the chlorophyll content of the wild-type plants decreased remarkably, whereas that of the nNOS-overexpressing rice plants remained relatively higher (Figure 3G). These results indicate that nNOS overexpression in rice can improve cadmium tolerance at the seedling stage.

### 3.3. nNOS-Overexpressing Rice Plants Accumulate Less Cadmium under Cadmium Stress

To determine whether the improved Cd tolerance of *nNOS*-overexpressing plants was associated with Cd accumulation in plant tissues, we transferred 2-day-old seedlings of both the wild-type and transgenic lines onto 1/2 MS medium with or without supplementation with 200 μM CdCl$_2$, and then the Cd contents of the shoot and roots were assayed 5 days after transfer. Under Cd stress, the Cd contents significantly decreased in both the roots and shoots of *nNOS* transgenic rice plants compared with wild-type plants (Figure 4A,B). Then, we used reverse-transcription quantitative PCR (RT-qPCR) to examine the expression levels of genes involved in Cd transport. Compared with the WT control, the expressions of the cadmium transporter genes *OsCAL1*, *OsIRT2*, *OsNramp5*, and *OsCd1* were significantly downregulated in the nNOS-overexpressing lines under cadmium treatment, whereas cadmium induced their expression (Figure 4C–F), suggesting that nitric oxide may reduce the accumulation of cadmium ions in rice seedlings by reducing the expressions of the genes encoding cadmium transporters.

### 3.4. ROS-Scavenging Capacity of nNOS-Overexpressing Rice Plants Was Enhanced

Much of the injury caused by cadmium at the cellular level is associated with oxidative damage due to ROS [1]. Additionally, treatment with the NO donor SNP can decrease ROS accumulation, thus alleviating oxidative damage in cadmium-stressed plants [7]. Therefore, we expected that our transgenic plants with higher NO contents would also repress cadmium-induced ROS accumulation, thereby achieving higher tolerance to Cd stress. For this purpose, we assayed the H$_2$O$_2$ contents of both wild-type and transgenic plants. The results showed that the transgenic plants repressed the H$_2$O$_2$ accumulation induced by cadmium stress in the wild-type plants; however, both wild-type and transgenic plants accumulated similar contents of H$_2$O$_2$ under normal conditions (Figure 5A). The reduced Cd-induced H$_2$O$_2$ content in transgenic plants may result from changes in the activities of antioxidant enzymes that can scavenge H$_2$O$_2$, such as CAT and POX. As expected, the transgenic plants showed much higher CAT and POX activities under Cd-

stress conditions compared with the wild-type plants (Figure 5B,C). Furthermore, qRT-PCR analysis showed that the expression levels of antioxidant enzymes genes such as *OsCATA*, *OsCATB*, and *OsPOX1* in the transgenic lines were significantly higher than those in wild-type plants when subjected to Cd stress (Figure 5D–F). These results suggest that endogenous NO improves Cd tolerance by increasing ROS-scavenging capacity under Cd stress.

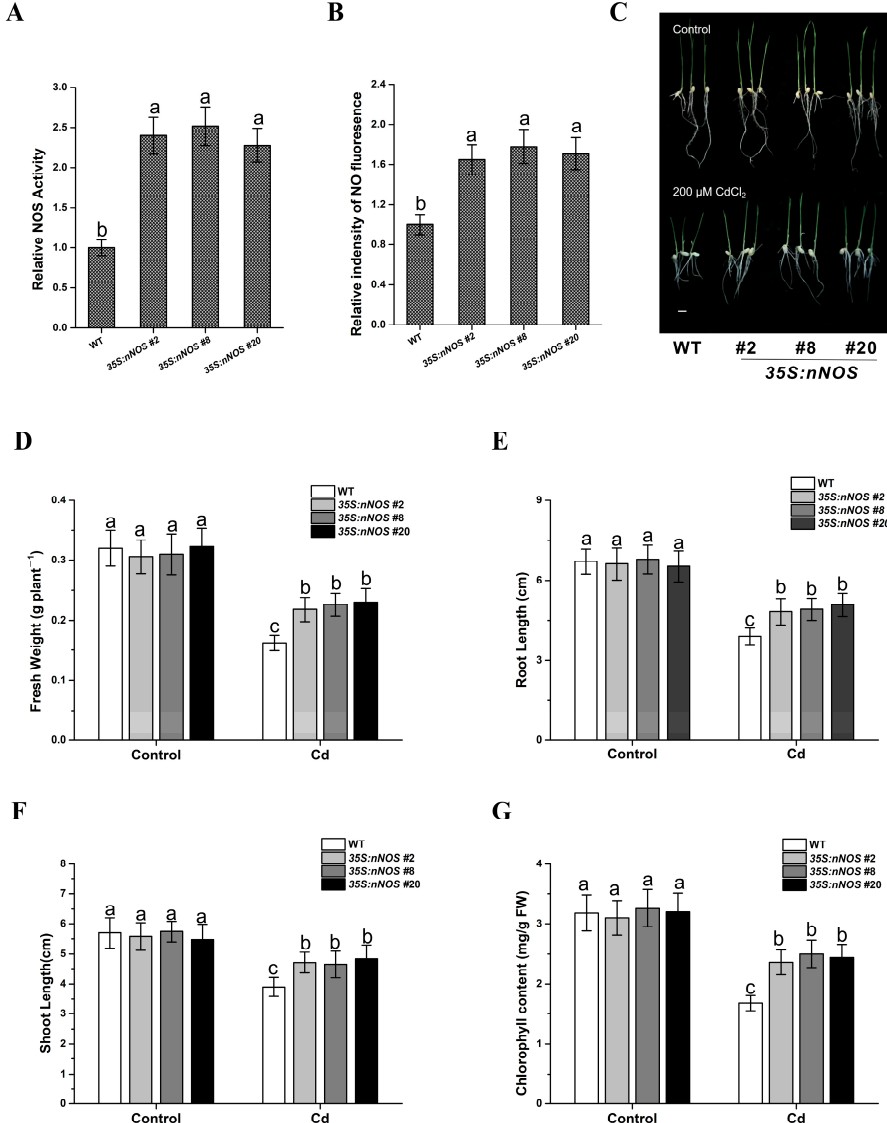

**Figure 3.** The *nNOS*-overexpressing lines showed enhanced cadmium tolerance. (**A**) NOS activities of the *nNOS*-overexpressing lines and wild-type plants were determined using a NOS assay kit. The relative NOS activity is expressed using the NOS activity of the wild type as the standard (1). (**B**) Relative NO content of the *nNOS*-overexpressing lines and wild-type plants examined using DAF-FM DA and expressed using the fluorescence of the wild-type plants as the standard (1). (**C**) Germinated wild-type and nNOS-overexpressing plants at 5 d after treatment with or without 200 μM CdCl$_2$. (**D–F**) The fresh weight (**D**), root length (**E**), shoot length (**F**), and total chlorophyll content (**G**) were assayed with both wild-type plants and three *nNOS*-overexpressing lines after 5 days of growth on the medium (1/2 MS with or without 200 μM CdCl$_2$). The results shown are the means ± SD. Values were derived from three independent biological experiments, and the different letters indicate significant differences between the annotated columns ($p < 0.05$ by Tukey's test). Bars = 1 cm; FW, fresh weight; Cd, 200 μM CdCl$_2$.

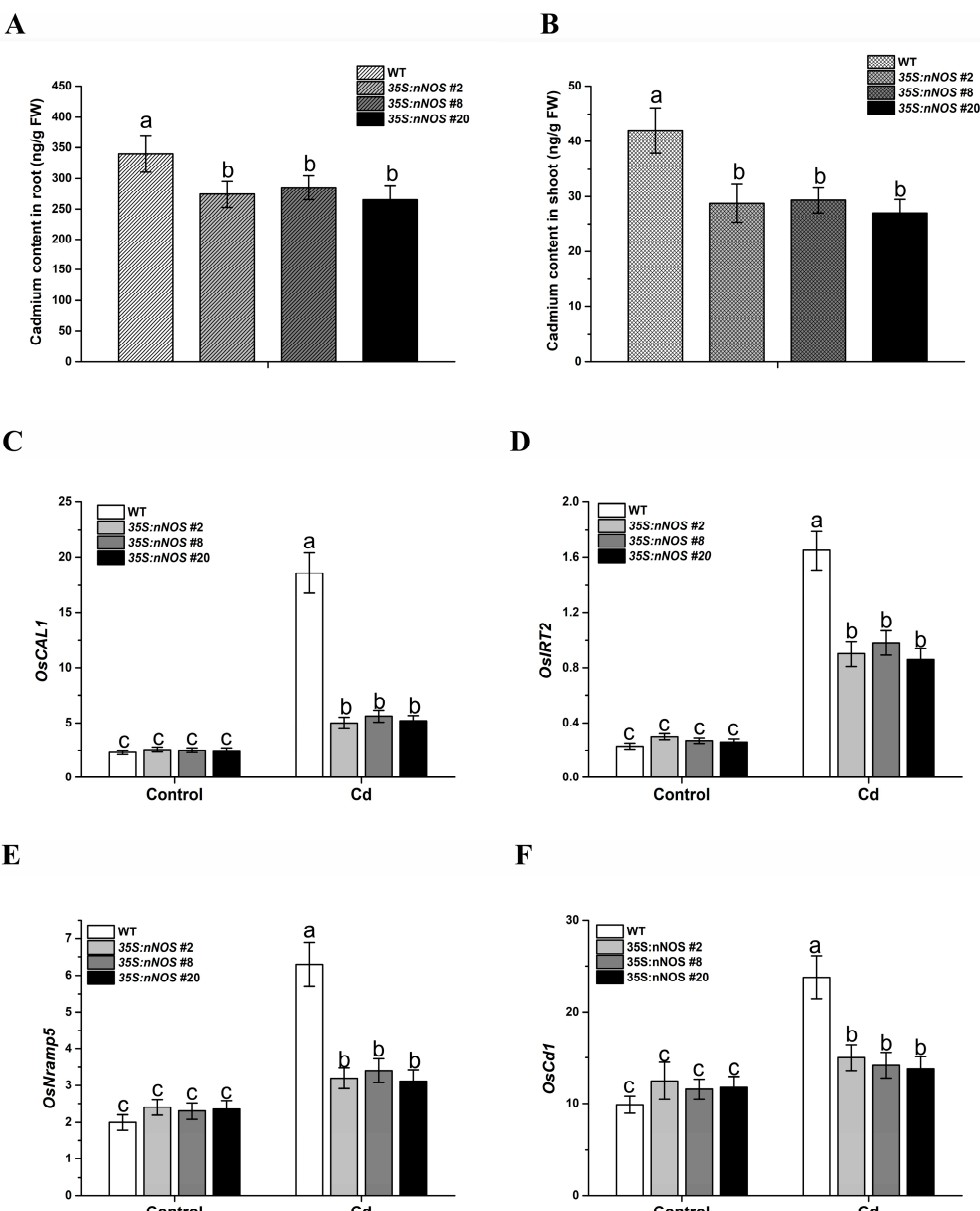

**Figure 4.** *nNOS*-overexpressing rice plants accumulated less cadmium under cadmium stress. Germinated wild-type and transgenic plants were treated with or without 200 μM CdCl$_2$ for 5 days, and then root (**A**) and shoot (**B**) cadmium contents, and the expression of *OsCAL1* (**C**), *OsIRT2* (**D**), *OsNramp5* (**E**), and *OsCd1* (**F**) were assayed. The results shown are the means ± SD. Values were derived from three independent biological experiments, and the different letters indicate significant differences between the annotated columns ($p < 0.05$ by Tukey's test). FW, fresh weight; Cd, 200 μM CdCl$_2$.

### 3.5. nNOS Transgenic Plants Showed Increased Melatonin Levels under Cadmium Stress

Many studies have reported that melatonin plays a similar role to NO in the tolerance of plants to Cd stress [24]. However, the relationship between NO and melatonin in response to Cd stress in plants is still unclear [37]. In this study, the melatonin content in *nNOS*-overexpressing transgenic lines was measured under cadmium stress. For this purpose, 2-day-old seedlings of both the wild-type and transgenic lines were transferred onto 1/2 MS medium with or without supplementation with 200 μM CdCl$_2$, and the melatonin contents were measured 5 d after transfer. As shown in Figure 6A,B, the melatonin levels significantly increased in the cadmium-treated plants compared with the untreated plants, and the melatonin contents in the nNOS-overexpressing transgenic lines

were markedly higher than those in the wild-type plants. The relative expression levels of the genes of the melatonin metabolism process, including *OsASMT1*, *OsTDC3*, *OsTDC1*, and *OsSNAT2*, increased under cadmium treatment in the transgenic lines compared with the WT lines (Figure 6C–F). All these results indicate that NO may reduce cadmium accumulation and improve cadmium tolerance in rice seedlings through modulation of melatonin biosynthesis.

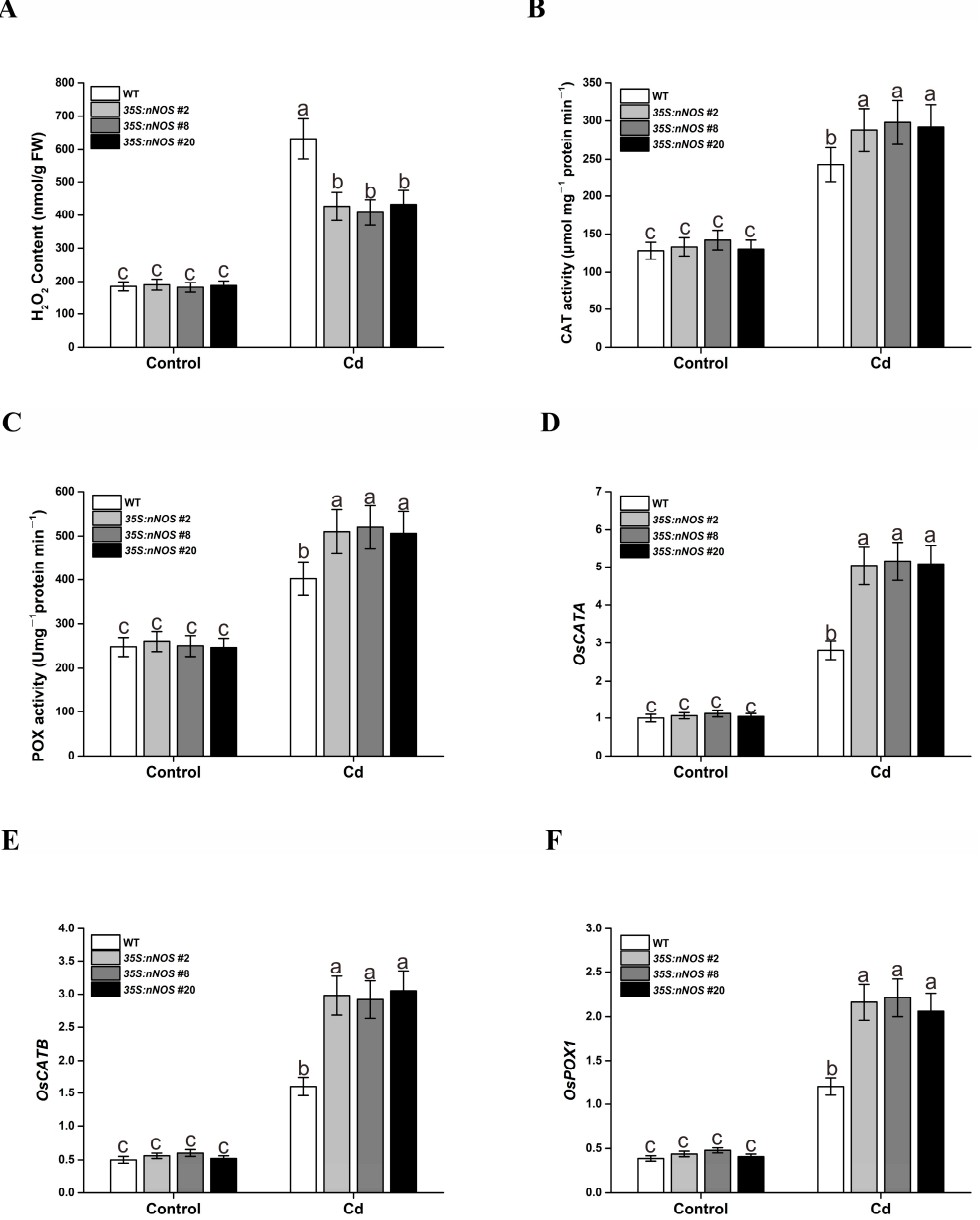

**Figure 5.** The *nNOS*-overexpressing lines exhibited improved ROS-scavenging capacity under cadmium stress. Germinated wild-type and *nNOS*-overexpressing plants were treated with or without 200 μM $CdCl_2$ for 5 days, and then $H_2O_2$ contents (**A**), enzymatic activities of CAT (**B**) and POX (**C**), and the expression of *OsCATA* (**D**), *OsCATB* (**E**) and *OsPOX1* (**F**) were assayed. The results shown are the means ± SD. Values were derived from three independent biological experiments, and the different letters indicate significant differences between the annotated columns ($p < 0.05$ by Tukey's test). FW, fresh weight; Cd, 200 μM $CdCl_2$.

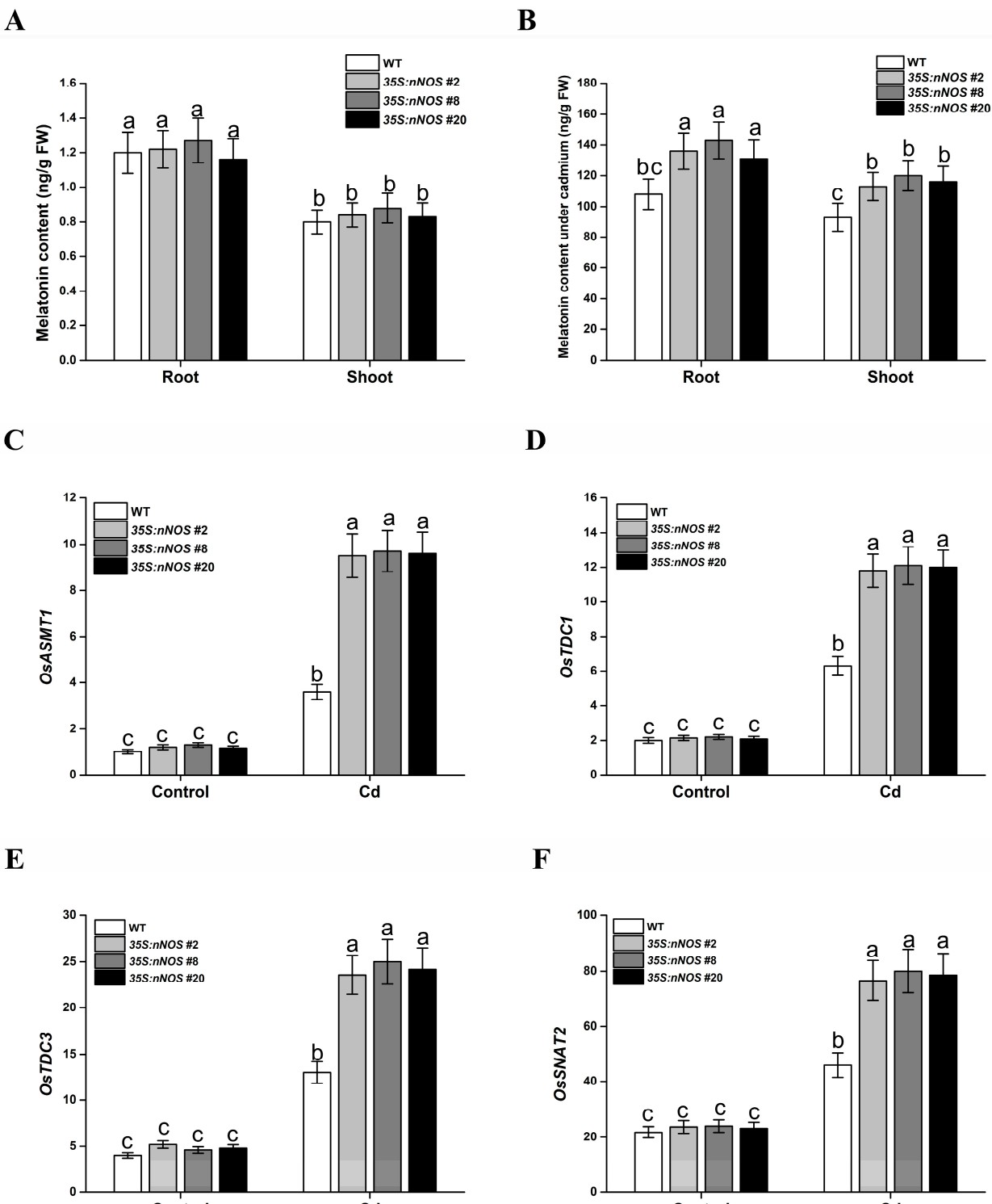

**Figure 6.** *nNOS*-overexpressing rice plants accumulate less cadmium under cadmium stress. Germinated wild-type and *nNOS*-overexpressing plants were treated with or without 200 μM CdCl$_2$ for 5 days, and then root (**A**) and shoot (**B**) melatonin contents, and the expression of *OsASMT1* (**C**), *OsTDC1* (**D**), *OsTDC3* (**E**), and *OsSNAT2* (**F**) were assayed. The results shown are the means ± SD. Values were derived from three independent biological experiments, and the different letters indicate significant differences between the annotated columns ($p < 0.05$ by Tukey's test). FW, fresh weight; Cd, 200 μM CdCl$_2$.

## 4. Discussion

Recently, many studies have reported the function of NO in response to Cd stress in plants. However, the role of NO in plant responses to Cd stress remains controversial [7]. This controversy may be due to the complex properties of NO, the functions of which depend on its location and concentration, as well as the plant species and plant development stage.

We present herein our results, showing that 200 μM CdCl$_2$ treatment induces NO accumulation in the roots of rice seedlings. This result is consistent with those of studies in which NO accumulation was enhanced after treatment with different concentrations of cadmium [11,13], although several studies have reported a decrease in NO content in rice upon cadmium treatment [10,15,38]. This difference may be due to the use of different Cd concentrations, treatment times, plant sizes, or genotypes. In addition, most results regarding NO function were achieved with the application of NO donors or scavengers, which may not adequately replicate the function of endogenous NO in plants. Therefore, an exploration of the roles of endogenous NO in response to Cd in rice also appears to be of great interest and importance. In this study, we also examined the effects of exogenous NO treatment on Cd resistance in rice seedlings, and further investigated the tolerance of *nNOS*-overexpressing rice plants with higher NOS activity and NO content to Cd stress. Consistent with the results of studies in which the application of an NO donor enhanced cadmium stress, the transgenic rice plants showed higher Cd tolerance and lower Cd accumulation.

When plants experience cadmium stress, the antioxidant system is activated to reduce the overproduction of ROS. Exogenous NO can alleviate Cd-induced oxidative damage by activating the antioxidant system and strengthening its ability to scavenge ROS [19,39]. NO can also scavenge excess ROS by directly binding to $O_2^-$ to form peroxynitrite ($ONOO^-$) [40]. Our results indicates that transgenic *nNOS* rice plants with higher NO levels accumulated less $H_2O_2$ under Cd stress conditions, possibly due to the upregulation of the expression of antioxidant enzyme genes (*OsCATA*, *OsCATB*, and *OsPOX1*). Whether the transgenic plants affect other ROS and how these antioxidant enzyme genes are regulated under Cd stress remains to be explored.

Exogenous application of an NO donor also affects the fixation of Cd and Cd uptake, thereby regulating the Cd tolerance and Cd accumulation of plants. For example, 0.1 mM SNP treatment increases the accumulation of Cd in the cell wall of roots but decreases Cd accumulation in the soluble fraction of leaves and roots in rice under Cd stress [15]. Exogenous NO can also reduce Cd uptake and transport in rice [19,39]. However, these studies mainly focus on the morphological and physiological indexes. In this study, *nNOS*-overexpressing rice plants accumulated less cadmium under Cd stress conditions, possibly due to the downregulation of the transcription of Cd uptake- and transport-related genes such as *OsCAL1*, *OsIRT2*, *OsNramp5*, and *OsCd1*.

Similar to the physiological functions of NO, melatonin alleviates the Cd-induced inhibition of seedling growth [24]. However, the relationship between NO and melatonin is still elusive. For example, exogenous melatonin improved Cd tolerance by reducing NO accumulation, resulting in lower Cd accumulation in *Brassica pekinensis* (Lour.) Rupr. [41], but triggered endogenous NO and alleviated Cd toxicity by increasing the activities of antioxidant enzymes in wheat seedlings [26]. These results suggest that NO may act as a downstream signal in response to Cd stress. However, Lee et al., in 2017, reported that cadmium-induced melatonin synthesis in rice requires NO [23]. In this paper, we present data showing that NO upregulates the expression of melatonin-biosynthesis-related genes, thus enhancing melatonin accumulation under cadmium stress. These findings suggest that NO may improve Cd tolerance by promoting melatonin biosynthesis. The complicated relationship between NO and melatonin requires further investigation.

Overall, this study showed the effects of exogenous and endogenous NO on the resistance of rice seedlings to Cd stress. NO was induced by Cd stress and exogenous NO donors were able to alleviate Cd toxicity in a dose-dependent manner. Rice plants

overexpressing rat *nNOS* showed increases in both NOS activity and NO content, resulting in improved Cd stress tolerance. Further analyses indicated that improving endogenous NO enhances cadmium tolerance in rice through modulation of cadmium accumulation, antioxidant capacity and melatonin biosynthesis (Figure 7). However, the manner in which NO modulates the expression of genes involved in these pathways and whether NO affects other pathways in order to alleviate Cd stress need to be further investigated.

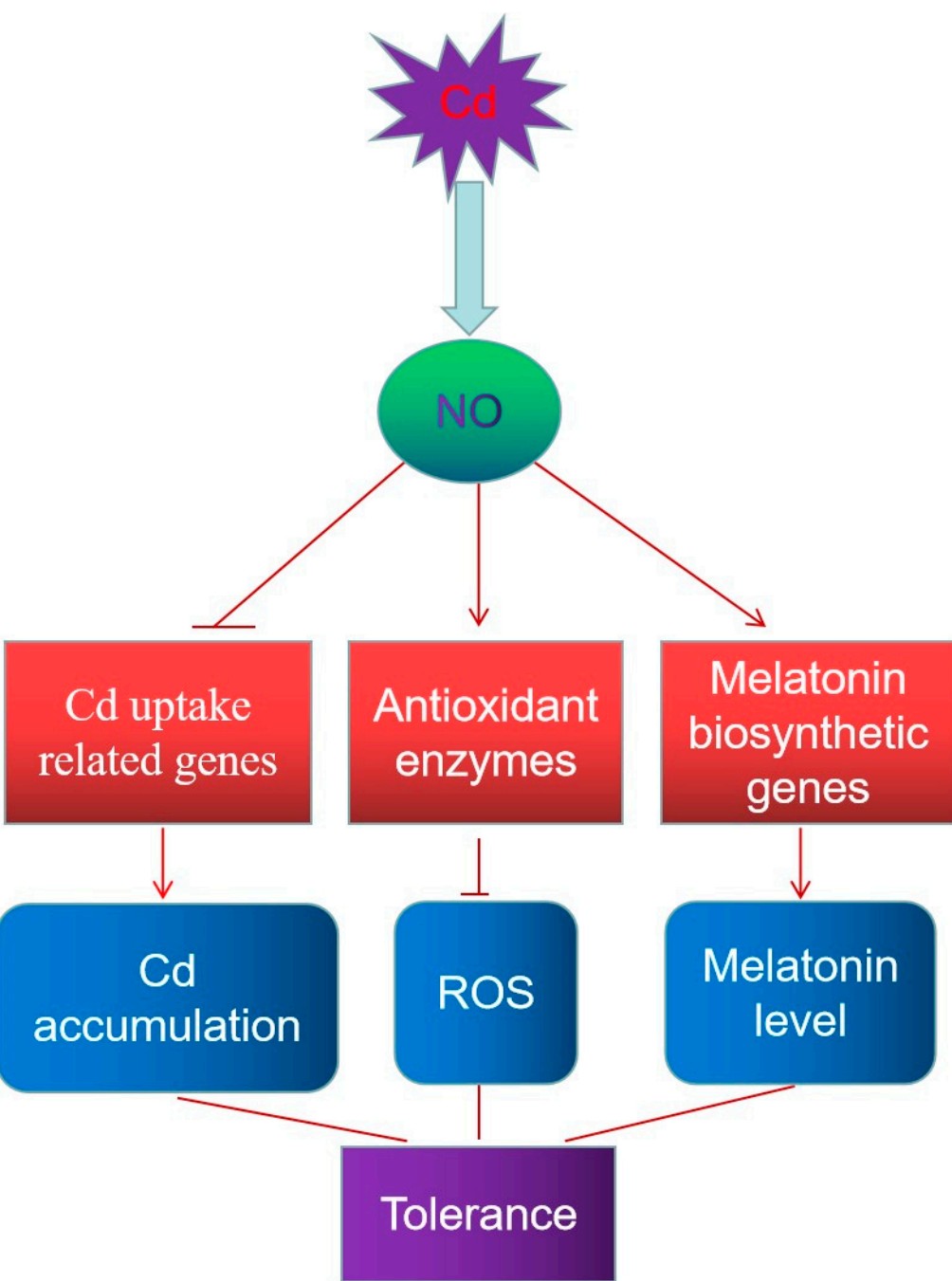

**Figure 7.** Model of the role of endogenous NO in Cd stress. Cd stress induces NO accumulation, and the generated NO enhances Cd tolerance through modulation of the contents of Cd, ROS, and melatonin, possibly by regulating the transcription level of genes involved in Cd uptake and antioxidant and melatonin biosynthesis.

## 5. Conclusions

When rice plants were stressed with Cd, the growth and development were significantly inhibited and excessive Cd was accumulated. The application of an exogenous NO donor alleviated Cd toxicity, and *nNOS*-overexpressing rice plants with higher NO levels showed improved Cd stress tolerance. Moreover, the transgenic rice plants accumulated less Cd, and had higher melatonin levels and stronger ROS-scavenging capacity, as well as regulating the transcription of related genes under Cd conditions to achieve higher tolerance to Cd stress. In summary, improving endogenous NO levels through exogenous application of SNP or transgenic technology can improve Cd tolerance and reduce Cd accumulation in rice seedlings.

**Supplementary Materials:** The following supporting information can be downloaded at: https://www.mdpi.com/article/10.3390/agronomy13081978/s1, Table S1. Primers used for qRT-PCR analysis. F: forward; R: reverse; Figure S1. Cadmium stress repress root and shoot growth of rice seedlings.

**Author Contributions:** Data curation, G.Z. and T.Y.; Formal analysis, W.C.; Investigation, H.D. and P.W.; Methodology, W.C., W.W., H.D., B.C., G.Z. and P.W.; Project administration, W.C. and Y.Z.; Resources, B.C.; Supervision, Y.Z.; Writing—original draft, W.C. and T.Y.; Writing—review and editing, W.W. and T.Y. All authors have read and agreed to the published version of the manuscript.

**Funding:** This study was mainly supported by the Hubei Provincial Natural Science Foundation (2019CFB399) and Knowledge Innovation Program of Wuhan-Basi Research (20200206601012298).

**Data Availability Statement:** The datasets generated and/or analyzed during the study are available from the senior author upon reasonable request.

**Conflicts of Interest:** The authors declare no conflict of interest.

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
