# Peer review of "Improving Endogenous Nitric Oxide Enhances Cadmium Tolerance in Rice through Modulation of Cadmium Accumulation and Antioxidant Capacity"

_agronomy, doi:10.3390/agronomy13081978_

Round 1
Reviewer 1 Report
Improving endogenous nitric oxide enhances cadmium tolerance in rice through modulation of cadmium accumulation and antioxidant capacity
The role of endogenous nitric oxide in enhancing the plant tolerance to Cd by rice is a common topic, which many studies already discussed this topic such as:
1- Long ZhangZhen ChenCheng Zhu (2012). Endogenous nitric oxide mediates alleviation of cadmium toxicity induced by calcium in rice seedlings. Journal of Environmental Sciences
2- Muhammad ImranSaddam HussainXiangru Tang (2023). Nitric oxide confers cadmium tolerance in fragrant rice by modulating physio-biochemical processes, yield attributes, and grain quality traits. Ecotoxicology and Environmental Safety
3- Jiuyue PanMeiyan GuanZhenzhen Cao (2021). Salicylic acid reduces cadmium (Cd) accumulation in rice (Oryza sativa L.) by regulating root cell wall composition via nitric oxide signaling. Science of The Total Environment
General comment:
Nearly all sections of MS contain a problem even in the not right writing or not correct writing. I did not find any section in this MS was written correctly????
Comments in details:
1- Abstract, this section is not accepted at all, where the right abstract, which includes main treatments, main findings in brief? Totally not corrected section?
2- Keywords, avoid any repeating of any word already mentioned in the title?
3- Introduction, based on the title, the authors must write 3 paragraphs (one on rice and its importance, one on cadmium and its potential, and one on NO, and finally maybe one on the link between all??
4- Materials and methods section is not clear? Flowchart for the main treatments, main measuring should be added?
How many days of this experiment? 2 or 5 days? It that enough? How did the authors make sure or guarantee their results within this very short period???
Is this period enough for uptake the tissues of root or shoot seedlings Cd???
5- Results: What is mentioned in Results section, should be moved into Materials section like:
“To elucidate how NO mediated cadmium stress in rice, we transferred 2-day-old wild-type (WT) rice seedlings germinated on half-strength Murashige and Skoog (1/2 MS) plates to new plates containing 0, 50, 100, or 200 μM CdCl2, and we measured the root and shoot lengths at 1, 2, 3, 4, and 5 days after transfer”
Is that only 5 days enough to get the roots and shoots of rice?? How? And how many mm will be in this case???
This part must move to Materials??? This is not results????
Again, this part, must move to the Materials section:
“Nitric oxide may dose-dependently regulate plant growth. We tested this hypothesis through experiments employing the exogenous application of the NO donor SNP. We transferred germinated rice seedlings to 1/2 MS medium containing 0, 20, 50, or 100 μM SNP with or without 200 μM CdCl2, and the root length and shoot length were measured 5 d after transfer”
What does mean SNP and other many abbreviations without mention the meaning????
6- Discussion, also not accepted
The authors must discuss their results not just review the published work related to their measurements
7- Conclusions also not accepted, this is not complete work?
8- The language is not so good and much concern on it is needed, as found in the MS like in abstract:
“In this study, rat neuron NO synthase (nNOS)-overexpressing rice plants,”
What does mean? Where the verb?
need more editing
Author Response
Dear reviewer:
We would like to thank you for your encouragement to submit the revised paper titled “Improving endogenous nitric oxide enhances cadmium tolerance in rice through modulation of cadmium accumulation and antioxidant capacity”. We also thank you for helpful suggestions to improve our manuscript. In this revised version, we have addressed all the issues raised by the reviewers. Below is a detailed description of the changes we have made in this revised version, along with our responses to each comments.
Improving endogenous nitric oxide enhances cadmium tolerance in rice through modulation of cadmium accumulation and antioxidant capacity
The role of endogenous nitric oxide in enhancing the plant tolerance to Cd by rice is a common topic, which many studies already discussed this topic such as:
- Long ZhangZhen, ChenCheng Zhu (2012). Endogenous nitric oxide mediates alleviation of cadmium toxicity induced by calcium in rice seedlings. Journal of Environmental Sciences
2- Muhammad Imran, Saddam Hussain,Xiangru Tang (2023). Nitric oxide confers cadmium tolerance in fragrant rice by modulating physio-biochemical processes, yield attributes, and grain quality traits. Ecotoxicology and Environmental Safety
3- Jiuyue Pan, Meiyan Guan, Zhenzhen Cao (2021). Salicylic acid reduces cadmium (Cd) accumulation in rice (Oryza sativa L.) by regulating root cell wall composition via nitric oxide signaling. Science of The Total Environment
Response: Many studies have reported the role of NO in response to Cd, but most results of NO functions are achieved by application of NO donors or scavengers, which may not replicate the function of endogenous NO in plants. Therefore, to explore the roles of endogenous NO in response to Cd in rice also appears to be of great interest and importance. Here, we performed our study with rice to address whether changes of NO level by overexpressing NOS could be involved in Cd stress response.
General comment:Nearly all sections of MS contain a problem even in the not right writing or not correct writing. I did not find any section in this MS was written correctly????
Response: Thanks for the suggestion. We have revised all sections of MS and hope that it would be suitable for publication.
Comments in details:
Point 1: Abstract, this section is not accepted at all, where the right abstract, which includes main treatments, main findings in brief? Totally not corrected section?
Response: Sorry for it. We have modified Abstract in this revised paper as follow: “
Nitric oxide (NO) plays an important role in plant stress responses. However, the mechanisms underlying NO-induced stress resistance to cadmium (Cd) stress in rice remain elusive. In this study, rat neuron NO synthase (nNOS)-overexpressing rice plants with higher endogenous NO level showed higher cadmium stress tolerance than the wild-type plants. The results showed that nNOS-overexpressing rice plants accumulated less cadmium in the roots and shoots by downregulating the expression of Cd uptake and transport related genes including OsCAL1, OsIRT2, OsNramp5, and OsCd1. Moreover, nNOS-overexpressing rice plants accumulated less hydrogen peroxide (H2O2), accompanying with higher expression of antioxidant enzyme genes (OsCATA, OsCATB, and OsPOX1) and corresponding higher enzyme activities under cadmium stress. Furthermore, the transcription of melatonin biosynthetic genes, including OsASMT1, OsTDC1, OsTDC3, and OsSNAT2, was also upregulated in nNOS-overexpressing plants, resulting in increased content of melatonin under cadmium treatment compared with the wild-type controls. Taken together, this study indicate that nNOS overexpression improves Cd tolerance of rice seedlings through decreasing cadmium accumulation and enhancing the antioxidant capacity and melatonin biosynthesis of the plants” Page 1
Point 2: Keywords, avoid any repeating of any word already mentioned in the title?
Response: We have modified Keywords in the revise version as follow:
“Keywords: nitric oxide; cadmium stress; reactive oxygen species; melatonin”Page 1
Point 3: Introduction, based on the title, the authors must write 3 paragraphs (one on rice and its importance, one on cadmium and its potential, and one on NO, and finally maybe one on the link between all??
Response: Thanks for your suggestion. To provide sufficient background, we have rewritten 2 paragraphs in the revised Introduction as follow:
“Rice is one of the most important crops in Asia. However, rice production safety is threatened by a a toxic heavy metal cadmium (Cd), due to the increasing Cd pollution problems. When rice plants grown in Cd-polluted soil, it can absorb Cd through roots and then transported to shoots and grains. The excessive accumulation of Cd in soil not only inhibits rice growth but also endangers human health through the food chain[1]. In order to resist Cd toxicity, rice has evolved many resistance strategies. On the one hand, rice reduce Cd accumulation by affecting Cd uptake ,transport and chelation. In recent years, many Cd uptake and transport related genes including OsCAL1(cadmium accumulation in leaf 1), OsIRT2(iron-regulated metal transporter2), OsNramp5(natural resistance-associated macrophage protein 5), and OsCd1(cadmium transporter gene 1) have been identified [2]. On the other hand, rice can reduce cell damage caused by Cd-induced overexpressed reactive oxygen species (ROS) through activating antioxidant enzymes such as superoxide dismutase (SOD), catalase (CAT) and peroxidase (POX)[1]. Thus, improving the cadmium tolerance and decreasing the cadmium accumulation in rice through biotechnology has become an urgent task given the increases in cadmium pollution problems.”Page 1-2
“Moreover, NO can also regulate hormone homeostasis such as indole-3-acetic acid (IAA) or melatonin (N-acetyl-5-methoxytryptamine), so as to alleviate Cd toxicity in plants [22-23]. Over past decades, increasing studies indicate that Cd stress can induce melatonin accumulation and exogenous melatonin improve Cd tolerance in different plants [24]. Melatonin is synthesized via four continual enzymatic reactions from tryptophan, requiring at least six enzymes: tryptophan decarboxylase (TDC), tryptophan hydroxylase (TPH), tryptamine 5-hydroxylase (T5H), N-acetylserotonin methyltransferase (ASMT), and serotonin N-acetyltransferase (SNAT)[24]. Many results showed that exogenous melatonin can alleviate Cd-induced oxidative damage by activating antioxidant systems [25-26], and decrease Cd accumulation by regulating the transcription of iron-transport genes [25,27]. Although melatonin and NO have similar role in response to Cd stress in plants, the relationship between them is still unclear.” Page 2
Point 4: Materials and methods section is not clear? Flowchart for the main treatments, main measuring should be added?How many days of this experiment? 2 or 5 days? It that enough? How did the authors make sure or guarantee their results within this very short period???Is this period enough for uptake the tissues of root or shoot seedlings Cd???
Response: We apologize for it. We added experimnets details of main treatments and main measuring in Material and Methods in this revised paper with following sentences:
2.2. Stress treatments and plant sampling
To evaluate a suitable Cd treatment concentration, we transferred 2-day-old wild-type (WT) rice seedlings germinated on half-strength Murashige and Skoog (1/2 MS) plates to new plates containing 0, 50, 100, or 200 μM CdCl2, and the seedlings were photographed and root and shoot lengths were measured using ImageJ software at 1, 2, 3, 4, and 5 days after transfer .
To test the effects of exogenous SNP treatment on the root and shoot growth under normal and cadmium conditions.We transferred germinated rice seedlings to 1/2 MS medium containing 0, 20, 50, or 100 μM SNP with or without 200 μM CdCl2, and the seedlings were photographed and root and shoot lengths were measured at 1, 2, 3, 4, and 5 days after transfer.
To evaluate the plants’ tolerance to cadmium stress, rice seeds were plated on agar medium containing 1/2 MS medium for 2 days in a plant growth chamber. Uniformly germinated rice seedlings were then transferred to 1/2 MS medium supplemented with 200 μM CdCl2. After 1, 2, 3, 4, and 5d of growth, the seedlings were photographed and root and shoot lengths were measured. At least 24 seedlings were analyzed per treatment.
To measure the transcript levels of selected genes and physiological parameters including H2O2 content, CAT and POX activity, chlorophyll content, cadmium content, and melatonin content under Cd stress, the roots, shoots, or whole seedlings of the tested plants in different treatment conditions were sampled at the designated time for further analyses. Page 3
2.3. Measurement of NO content
1-week-old wild-type rice seedlings were treated in 200 μM CdCl2 and the NO content in the roots was assayed using the specific fluorescent probe DAF-FM DA at 24 hours post-treatment[33]. Page 3
2.6. Quantitative real-time PCR
Rice seedlings treated with or without 200 μM CdCl2 for 5 days were sampled for qRT-PCR as previously described [33]. Page 4
2.7. Measurement of Cd Content
The Cd content was analysed according to a method described previously [34]. Briefly, rice seedlings treated with or without 200 μM CdCl2 for 5 days and root and shoot were sampled, then the samples were washed with sterile water and then dried at 80°C for 1d. Page 4
2.8. Measurement of Chlorophyll Content
The chlorophyll content was determined according to a previously described method [35]. Briefly, 0.5g fresh leaves of plants from different lines which treated with or without 200 μM CdCl2 for 5 days were collected and incubated in 20 mL of 80% (v/v) acetone and kept in darkness for 24h. After centrifugation, the extracted solutions were used for the total chlorophyll content determination. Page 4
2.9. Measurement of melatonin content
Melatonin content in rice tissues was determined as described previously [36]. Briefly, after treatment with or without 200μM CdCl2 for 5 days, root and shoot samples were extracted with acetone:methanol : water (v : v : v) = 89 : 10 : 1) and centrifuged. Subsequently, the supernatant was used for the melatonin content determination by using a melatonin enzyme-linked immunosorbent assay (ELISA) kit. Page 5
As we mentioned in MM, rice seeds grown on 1/2 MS for 2 days and germinated seedlings treated with or without 200 μM CdCl2 for 5 days, then the root and shoot length, and the transcript levels of selected genes and physiological parameters were measured. The results showed that the root and shoot growth were markedly inhibited by 5-day 200μM CdCl2 treatment. We also found the Cd contents were significantly increased in both the roots and shoots of rice plants compared with those in untreated controls (Figure 4A,B).
Point 5-1: Results: What is mentioned in Results section, should be moved into Materials section like:“To elucidate how NO mediated cadmium stress in rice, we transferred 2-day-old wild-type (WT) rice seedlings germinated on half-strength Murashige and Skoog (1/2 MS) plates to new plates containing 0, 50, 100, or 200 μM CdCl2, and we measured the root and shoot lengths at 1, 2, 3, 4, and 5 days after transfer”. Is that only 5 days enough to get the roots and shoots of rice?? How? And how many mm will be in this case???This part must move to Materials??? This is not results????
Response: Thanks for the suggestion. We have rewritten the related sentences as follow: “To elucidate how NO mediated cadmium stress in rice, we examined the root and shoot growth in different Cd concentrations. As shown in Supplementary Figure 1A-B , the root and shoot length were markedly inhibited by 200 μM CdCl2 compared with those in untreated control. Therefore, 200 μM CdCl2 was used in the subsequent experiments based on the above results.” Page 5
We also added the details of this treatment in Materials and Methods as follow: “To evaluate a suitable Cd treatment concentration, we transferred 2-day-old wild-type (WT) rice seedlings germinated on half-strength Murashige and Skoog (1/2 MS) plates to new plates containing 0, 50, 100, or 200 μM CdCl2, and the seedlings were photographed and root and shoot lengths were measured using ImageJ software at 1, 2, 3, 4, and 5 days after transfer.”Page 3
This experiment is considered to explore an appropriate cadmium treatment concentration. As shown in Supplementary Figure 1A-B, the root and shoot length were markedly inhibited by 200 μM CdCl2 compared with those in untreated control. Therefore, 200 μM CdCl2 was used in the subsequent experiments based on the above results .
Point 5-2: Again, this part, must move to the Materials section:“Nitric oxide may dose-dependently regulate plant growth. We tested this hypothesis through experiments employing the exogenous application of the NO donor SNP. We transferred germinated rice seedlings to 1/2 MS medium containing 0, 20, 50, or 100 μM SNP with or without 200 μM CdCl2, and the root length and shoot length were measured 5 d after transfer”.
Response: Thanks for the suggestion. We have rewritten the related sentences as follow: “Nitric oxide may dose-dependently regulate plant growth. We tested this hypothesis through experiments employing the exogenous application of different concentrations of NO donor SNP. ” Page 5
We also described this treatment in Materials and Methods as follow:
“To test the effects of exogenous SNP treatment on the root and shoot growth under normal and cadmium conditions.We transferred germinated rice seedlings to 1/2 MS medium containing 0, 20, 50, or 100 μM SNP with or without 200 μM CdCl2, and the seedlings were photographed and root and shoot lengths were measured at 1, 2, 3, 4, and 5 days after transfer.”Page 3
Point 5-3: What does mean SNP and other many abbreviations without mention the meaning????
Response: We have included SNP in Introduction as follow: “For instance, inhibition of NO accumulation by NO scavenger 2-[4-carboxyphenyl]-4,4,5,5-tetramethylimidazoline-1-oxy-3-oxide (c-PTIO) or NOS inhibitor Nω-nitro-L-arginine-methylester (L-NAME) result in prevention of Cd stress induced oxidant damage in Arabidopsis and yellow lupine [16-17], but application of a NO donor sodium nitroprusside (SNP) decreases ROS accumulation in Cd stressed Brassica juncca and rice seedlings [18-19].”Page 2
We also include other abbreviations that first appeared in this paper as follow:
“ In recent years, many Cd uptake and transport related genes including OsCAL1(cadmium accumulation in leaf 1), OsIRT2 (iron-regulated metal transporter 2), OsNramp5 (natural resistance-associated macrophage protein 5), and OsCd1(cadmium transporter gene 1) have been identified [2]. On the other hand, rice can reduce cell damage caused by Cd-induced overexpressed reactive oxygen species (ROS) through activating antioxidant enzymes such as superoxide dismutase (SOD), catalase (CAT) and peroxidase (POX)[1].”Page 2
“Melatonin is synthesized via four continual enzymatic reactions from tryptophan, requiring at least six enzymes: tryptophan decarboxylase (TDC), tryptophan hydroxylase (TPH), tryptamine 5-hydroxylase (T5H), N-acetylserotonin methyltransferase (ASMT), and serotonin N-acetyltransferase (SNAT) [24].”Page 2
Point 6: Discussion, also not accepted. The authors must discuss their results not just review the published work related to their measurements.
Response: Thanks for the suggestion. We have rewritten Discussion in the revised paper as follow:
Recently, many studies have reported the function of NO in response to Cd stress in plants. However, the role of NO in plant responses to Cd stress remains controversial [7]. This controversy maybe due to the complex properties of NO, which functions depend on its location and concentration, plant species, as well as plant development stages.
We present herein our results showing that 200 μM CdCl2 treatment induces NO accumulation in the root of rice seedlings. This result is consistent with those of studies in which NO accumulation was enhanced after treatment with different concentrations of cadmium [13,38], though several studies have reported that NO content in rice was decreased by cadmium treatment [10,15,39]. This difference may be due to different Cd concentration, treatment time, plant size, and genotype. In addition, most results of NO functions are achieved by application of NO donors or scavengers, which may not replicate the function of endogenous NO in plants. Therefore, to explore the roles of endogenous NO in response to Cd in rice also appears to be of great interest and importance. In this study, we also examined the effects of exogenous NO treatment on Cd resistance in rice seedlings, and further investigated the tolerance of nNOS-overexpressing rice plants with higher NOS activity and NO content to Cd stress. Consistent with those of studies in which the application of a NO donor enhanced cadmium stress, the transgenic rice plants showed higher Cd tolerance and lower Cd accumulation.
When plants are stressed by cadmium, the antioxidant system would be activated to reduce the overproduced ROS. Exogenous NO can alleviate the Cd-induced oxidative damage by activating the antioxidant system and strengthening its ability to scavenge ROS [40-41]. NO can also scavenge excess ROS by directly binding to O2- to form peroxynitrite (ONOO−) [42]. Our results indicated that transgenic nNOS rice plants which had higher NO level accumulated less H2O2 under Cd stress conditions, possibly by upregulating the expression of antioxidant enzyme genes (OsCATA, OsCATB, and OsPOX1). Whether the transgenic plants affect other ROS and how these antioxidant enzyme gene are regulated under Cd stress remain to be explored.
Exogenous application of NO donor also affects the fixation of Cd and Cd uptake, thus regulating the Cd tolerance and Cd accumulation of plants. For example, 0.1mM SNP treatment increases the accumulation of Cd in the cell wall of root but decrease Cd accumulation in the soluble fraction of leaves and roots in rice under Cd stress [15]. Exogenous NO can also reduce Cd uptake and transport in rice [40-41]. However, these studies mainly focus on the morphological and physiological indexes. In this study, the nNOS-overexpressing rice plants accumulated less cadmium under Cd stress conditions, possibly by downregulating the transcription of Cd uptake and transport related genes such as OsCAL1, OsIRT2, OsNramp5, and OsCd1.
Similar to the physiological functions of NO, melatonin alleviates the Cd-induced inhibition of seedling growth [24]. However, the relationship between NO and melatonin is still elusive. For example, exogenous melatonin improved Cd tolerance by reducing NO accumulation resulting in lower Cd accumulation in Brassica pekinensis (Lour.) Rupr. [43], but triggered endogenous NO and alleviate Cd toxicity via increasing the activities of antioxidant enzymes in wheat seedlings [26].These results suggest that NO may act as downstream signal in response to Cd stress. However, Lee et al., (2017) reported that cadmium-induced melatonin synthesis in rice requires nitric oxide. In this paper, we present data showing that NO up-regulate the expression of melatonin biosynthesis related genes, thus enhancing melatonin accumulation under cadmium stress. This findings suggest that NO may improve Cd tolerance through promoting melatonin biosynthesis. The complicated relationship between NO and melatonin requires further investigation.
Overall, this study showed the effects of exogenous and endogenous NO on the resistance of rice seedlings to Cd stress. NO is induced by Cd stress and exogenous NO donor can alleviate Cd toxicity in a does-dependent manner. Rice plants overexpressing rat nNOS showed increases in both the NOS activity and NO content , resulting in improved Cd stress tolerance. Further analyses indicated that improving endogenous NO enhances cadmium tolerance in rice through modulation of cadmium accumulation, antioxidant capacity and melatonin biosynthesis (Figure 7). However, how NO modulates the expression of genes involved in these pathways and whether NO affects other pathways to alleviate Cd stress need to be further investigated. Page 16-17
Point 7: Conclusions also not accepted, this is not complete work?
Response: Thanks for the suggestion. We have rewritten Conclusions in the revised paper as follow:“When rice plants stressed by Cd, the growth and development were significantly inhibited and excessive Cd accumulated. The application of exogenous NO donor alleviated Cd toxicity and nNOS-overexpressing rice plants with higher NO level showed improved Cd stress tolerance. Moreover, the transgenic rice plants accumulated less Cd, had higher melatonin levels and stronger ROS-scavenging capacity, as well as regulate transcription of related genes under Cd conditions for higher tolerance to Cd stress. In summary, improving endogenous NO level through exogenous application of SNP or transgenic technology can improve Cd tolerance and reduce Cd accumulation in rice seedlings.”Page 18
Point 8: The language is not so good and much concern on it is needed, as found in the MS like in abstract:“In this study, rat neuron NO synthase (nNOS)-overexpressing rice plants,”What does mean? Where the verb?
Response: Sorry for it. We have modified this sentence in the revised version as follow:“In this study, rat neuron NO synthase (nNOS)-overexpressing rice plants with higher endogenous NO level showed higher cadmium stress tolerance than the wild-type plants.”
Page 1
Point 9: Comments on the Quality of English Language: need more editing
Response:We have carefully edited the paper at our best and we also edited MS in MDPI before we submitted it to Agronomy.

Reviewer 2 Report
The increasing interest on cadmium (Cd) toxicity is because of its harmful effects on plant growth as well as potential health risks connected with food chain pollution. Nitric oxide (NO) is a bioactive molecule in plants which mediates a variety of physiological processes and responses to biotic and abiotic stresses including heavy metals. However, the role of NO in the response of plants to Cd stress remains ambiguous. In the present study, the mechanisms underlying NO-induced stress resistance to cadmium (Cd) in rice was postulated. Wild-type rice seedlings, as well as rice seedlings overexpressing rat neuron NO synthase (nNOS), which exhibited higher endogenous NOS activity and NO content, were used as model plants. It was evidenced that endogenous nitric oxide enhances cadmium tolerance in rice through modulation of cadmium accumulation and antioxidant capacity.
Comments:
Materials and methods
Page 2-3
2. 2.1. Plant materials and growth conditions
Rice (Oryza sativa L. cv. Zhonghua11) was used as the wild-type and in generation of nNOS transgenic plants. Rice seeds were sterilized with 70% (v/v) ethanol for 5 min and subsequently with 5% (w/v) NaClO for 30 min, washed at least three times with sterile water, and then plated on agar medium containing 1/2 MS medium in plant growth chambers (50% humidity, 200 μmol m –2 s –1 , 14 h light/10 h dark cycle, and 28-30°C).
2.2. Stress treatments and plant sampling
To evaluate the plants’ tolerance to cadmium stress, rice seeds were sterilized with 70% (v/v) ethanol for 5 min and subsequently with 5% (w/v) NaClO for 30 min, washed at least three times with sterile water, and then plated on agar medium containing 1/2 MS medium for 2 days in a plant growth chamber.
These parts of the Materials and Methods comprise actually the same information, therefore they should be redrafted and shortened.
Page 3
2.5. Measurement of H2O2 content, CAT activity and POX activity
What is the meaning of the abbreviations CAT and POX? What is the function of these proteins in removing of reactive oxygen species? This information should be included in Introduction.
Results
Page 5
Figure 2. Exogenous NO donor alleviated Cd toxicity in rice seedlings. (A) Effect of cadmium stress on the endogenous NO level in the roots of wild-type rice plants. 1-week-old wild-type plants were treated with 200 μM CdCl2 combined with or without 1 mM L-NAME for 24 h, and then NO production was measured. NO fluorescence in roots of the nNOS-overexpressing lines and wild-type plants examined using DAF-FM DA . (B and C) Effects of SNP or L-NAME on the root (B) and shoot (C) growth under cadmium stress. The results shown are the mean ± SD. Values are derived from three independent biological experiments. Different letters indicate significant differences using Tukey’s multiple comparison test at p<0.05. Bars = 50 μm; Cd, 200 μM CdCl 2; SNP, 20 μM SNP; L-NAME, 1 mM L-NAME.
The general caption should be changed due to it is confusing.
Exogenous NO donor alleviated Cd toxicity in rice seedlings.
This sentence relates only to graphs B and C.
Graph A represents effect of cadmium stress on the endogenous NO level in the roots of wild-type rice plants which were not treated with SNP as an exogenous donor of NO.
Page 5
Figure 2.
1-week-old wild-type plants were treated with 200 μM CdCl2 combined with or without 1 mM L-NAME for 24 h, and then NO production was measured.
This sentence is not clear. Which interpretation is correct?
1. Wild-type plants were treated with 200 μM CdCl2 for 24 h (with or without 1 mM L-NAME). Why were they not exposed for 5 days to 200 μM CdCl2 ?
2. 2-day-old wild-type seedlings were transferred onto medium supplemented with or without 200 μM CdCl2 for 5 days and subsequently exposed to combination of 200 μM CdCl2 with 1 mM L-NAME for 24 h. Control plants were treated for further 24 h only with 200 μM CdCl2.
Page 8
Figure 3.
(C) Germinated wild-type and nNOS-overexpressing plants were treated with or without 200 μM CdCl2, and the appearance of 200 μM CdCl2 treated wild-type plants and nNOS-overexpressing at 5 d after treatment.
This caption should be revised.
Suggestion: Germinated wild-type and nNOS-overexpressing plants at 5 d after treatment with or without 200 μM CdCl2
Page 9
3.3. nNOS-overexpressing rice plants accumulate less cadmium under cadmium stress
Then, we use quantitative real-time PCR (qRT-PCR) to examine the expression levels of several Cd uptake and transport related genes. Compared with the WT control, the expressions of OsCAL1, OsIRT2, OsNramp5, and OsCd1 were significantly downregulated in the nNOS overexpressing lines under cadmium treatment, whereas cadmium induced their expression (Figure 4C–F).
What do the abbreviations OsCAL1, OsIRT2, OsNramp5 and OsCd1 mean? What are the functions of the proteins encoded by these genes? This information should be included in Introduction.
Page 11
3.4. ROS-scavenging capacity of nNOS-overexpressing rice plants was enhanced
Much of the injury caused by cadmium stress at the cellular level is due to overproduced ROS [1]. Additionally, treatment with the NO donor SNP can decrease ROS accumulation, thus alleviating oxidative damage in cadmium-stressed plants [6]. Therefore, whether our nNOS-overexpressing plants with enhanced Cd tolerance also overproduced ROS [1].
Results and conclusions from previously published reports should be included in Introduction.
Page 13
3.5. nNOS transgenic plants showed increased melatonin levels under cadmium stress
Many studies have reported that melatonin plays a similar role to NO in tolerance of plants to Cd stress [32]. However, the relationship between NO and melatonin in response to Cd stress in plants is still unclear [33].
Results and conclusions from previously published reports should be included in Introduction.
Page 13
3.5. nNOS transgenic plants showed increased melatonin levels under cadmium stress
The relative expression levels of the genes of the melatonin metabolism process, including OsASMT1, OsTDC3, OsTDC1, and OsSNAT2, were increased under cadmium treatment in the transgenic lines compared with the WT lines (Figure 6C–F).
What do the abbreviations OsASMT1, OsTDC3, OsTDC1, and OsSNAT2 mean? What are the functions of the proteins encoded by these genes? This information should be included in Introduction.
Discussion
Page 15
Recently, Many studies have reported the function of NO in response to Cd stress in plants. However, the role of NO in plant responses to Cd stress are still debated [6].
For example, Exogenous melatonin improved Cd tolerance by reducing NO accumulation resulting in lower Cd accumulation in Brassica pekinensis (Lour.)
Capital letter instead of lowercase
Author Response
Dear reviewer:
We would like to thank you for your encouragement to submit the revised paper titled “Improving endogenous nitric oxide enhances cadmium tolerance in rice through modulation of cadmium accumulation and antioxidant capacity”. We also thank you for helpful suggestions to improve our manuscript. In this revised version, we have addressed all the issues raised by the reviewers. Below is a detailed description of the changes we have made in this revised version, along with our responses to each comments.
Comments and Suggestions for Authors
The increasing interest on cadmium (Cd) toxicity is because of its harmful effects on plant growth as well as potential health risks connected with food chain pollution. Nitric oxide (NO) is a bioactive molecule in plants which mediates a variety of physiological processes and responses to biotic and abiotic stresses including heavy metals. However, the role of NO in the response of plants to Cd stress remains ambiguous. In the present study, the mechanisms underlying NO-induced stress resistance to cadmium (Cd) in rice was postulated. Wild-type rice seedlings, as well as rice seedlings overexpressing rat neuron NO synthase (nNOS), which exhibited higher endogenous NOS activity and NO content, were used as model plants. It was evidenced that endogenous nitric oxide enhances cadmium tolerance in rice through modulation of cadmium accumulation and antioxidant capacity.
Comments:
Materials and methods
Point1: Page 2-3
2.1. Plant materials and growth conditions
Rice (Oryza sativa L. cv. Zhonghua11) was used as the wild-type and in generation of nNOS transgenic plants. Rice seeds were sterilized with 70% (v/v) ethanol for 5 min and subsequently with 5% (w/v) NaClO for 30 min, washed at least three times with sterile water, and then plated on agar medium containing 1/2 MS medium in plant growth chambers (50% humidity, 200 μmol m –2 s –1 , 14 h light/10 h dark cycle, and 28-30°C).
2.2. Stress treatments and plant sampling
To evaluate the plants’ tolerance to cadmium stress, rice seeds were sterilized with 70% (v/v) ethanol for 5 min and subsequently with 5% (w/v) NaClO for 30 min, washed at least three times with sterile water, and then plated on agar medium containing 1/2 MS medium for 2 days in a plant growth chamber.
These parts of the Materials and Methods comprise actually the same information, therefore they should be redrafted and shortened.
Response: Thanks for the suggestion. We have shortened this part in 2.2. Stress treatments and plant sampling as follow: “To evaluate the plants’ tolerance to cadmium stress, rice seeds were plated on agar medium containing 1/2 MS medium for 2 days in a plant growth chamber. ” Page 3
Point 2: Page 3
2.5. Measurement of H2O2 content, CAT activity and POX activity
What is the meaning of the abbreviations CAT and POX? What is the function of these proteins in removing of reactive oxygen species? This information should be included in Introduction.
Response: Thanks for the suggestion. We have included CAT and POX in revised Introduction as follow: “On the other hand, rice can reduce cell damage caused by Cd-induced overexpressed reactive oxygen species (ROS) through activating antioxidant enzymes such as superoxide dismutase (SOD), catalase (CAT) and peroxidase (POX)[1].”Page 3
Results
Point 3: Page 5
Figure 2. Exogenous NO donor alleviated Cd toxicity in rice seedlings. (A) Effect of cadmium stress on the endogenous NO level in the roots of wild-type rice plants. 1-week-old wild-type plants were treated with 200 μM CdCl2 combined with or without 1 mM L-NAME for 24 h, and then NO production was measured. NO fluorescence in roots of the nNOS-overexpressing lines and wild-type plants examined using DAF-FM DA . (B and C) Effects of SNP or L-NAME on the root (B) and shoot (C) growth under cadmium stress. The results shown are the mean ± SD. Values are derived from three independent biological experiments. Different letters indicate significant differences using Tukey’s multiple comparison test at p<0.05. Bars = 50 μm; Cd, 200 μM CdCl 2; SNP, 20 μM SNP; L-NAME, 1 mM L-NAME.
The general caption should be changed due to it is confusing. Exogenous NO donor alleviated Cd toxicity in rice seedlings.This sentence relates only to graphs B and C. Graph A represents effect of cadmium stress on the endogenous NO level in the roots of wild-type rice plants which were not treated with SNP as an exogenous donor of NO.
Response: Sorry. We modified the title of Figure 2 in revised version as follow: “Endogenous NO level in response to Cd stress and the effect of exogenous NO donor on Cd toxicity in rice seedlings.”Page 7
Point 4: Page 5
Figure 2.
1-week-old wild-type plants were treated with 200 μM CdCl2 combined with or without 1 mM L-NAME for 24 h, and then NO production was measured.
This sentence is not clear. Which interpretation is correct?
- Wild-type plants were treated with 200 μM CdCl2for 24 h (with or without 1 mM L-NAME). Why were they not exposed for 5 days to 200 μM CdCl2 ?
- 2-day-old wild-type seedlings were transferred onto medium supplemented with or without 200 μM CdCl2for 5 days and subsequently exposed to combination of 200 μM CdCl2 with 1 mM L-NAME for 24 h. Control plants were treated for further 24 h only with 200 μM CdCl2.
Response: Sorry for it. Wild-type plants were treated with 200 μM CdCl2 for 24 h (with or without 1 mM L-NAME), should be the correct interpretation. We also examined the NO content when it exposed to 200 μM CdCl2 for 5 days.The results showed that NO fluorescence also increased compared to those in untreated control, but the difference was not as significant as 24-hour treatment.
Point 5: Page 8
Figure 3 (C) Germinated wild-type and nNOS-overexpressing plants were treated with or without 200 μM CdCl2, and the appearance of 200 μM CdCl2 treated wild-type plants and nNOS-overexpressing at 5 d after treatment.
This caption should be revised.
Suggestion: Germinated wild-type and nNOS-overexpressing plants at 5 d after treatment with or without 200 μM CdCl2.
Response: Thanks for the suggestion.We have modified this sentence in the revised version as follow: “Germinated wild-type and nNOS-overexpressing plants at 5 d after treatment with or without 200 μM CdCl2.”Page 9. Figure 3(C)
Point 6: Page 9
3.3. nNOS-overexpressing rice plants accumulate less cadmium under cadmium stress
Then, we use quantitative real-time PCR (qRT-PCR) to examine the expression levels of several Cd uptake and transport related genes. Compared with the WT control, the expressions of OsCAL1, OsIRT2, OsNramp5, and OsCd1 were significantly downregulated in the nNOS overexpressing lines under cadmium treatment, whereas cadmium induced their expression (Figure 4C–F).
What do the abbreviations OsCAL1, OsIRT2, OsNramp5 and OsCd1 mean? What are the functions of the proteins encoded by these genes? This information should be included in Introduction.
Response: Thanks for the suggestion. We have included OsCAL1, OsIRT2, OsNramp5 and OsCd1 in revised Introduction as follow: “In order to resist Cd toxicity, rice has evolved many resistance strategies. On the one hand, rice reduce Cd accumulation by affecting Cd uptake, transport and chelation. In recent years, many Cd uptake and transport related genes including OsCAL1(cadmium accumulation in leaf 1), OsIRT2(iron-regulated metal transporter2), OsNramp5(natural resistance-associated macrophage protein 5), and OsCd1(cadmium transporter gene 1)have been identified [2].”Page 2
Point 7: Page 11
3.4. ROS-scavenging capacity of nNOS-overexpressing rice plants was enhanced
Much of the injury caused by cadmium stress at the cellular level is due to overproduced ROS [1]. Additionally, treatment with the NO donor SNP can decrease ROS accumulation, thus alleviating oxidative damage in cadmium-stressed plants [6]. Therefore, whether our nNOS-overexpressing plants with enhanced Cd tolerance also overproduced ROS [1].
Results and conclusions from previously published reports should be included in Introduction.
Response: Thanks for the suggestion. Results and conclusions mentioned in this part we already included in Introduction as follow:
“On the other hand, rice can reduce cell damage caused by Cd-induced overexpressed reactive oxygen species (ROS) through activating antioxidant enzymes such as superoxide dismutase (SOD), catalase (CAT) and peroxidase (POX)[1].” Page 2, first paragraph
and “Recently, researchers have been increasingly reporting that NO modulate resistance of plants to Cd stress by affecting physiological metabolic processes such as reactive oxygen species (ROS), photosynthesis, chlorophyll synthesis, and cadmium uptake [6-8], but the role of NO in response to Cd stress is still disputed.” and “For instance, inhibition of NO accumulation by NO scavenger 2-[4-carboxyphenyl]-4,4,5,5-tetramethylimidazoline-1-oxy-3-oxide (c-PTIO) or NOS inhibitor Nω-nitro-L-arginine-methylester(L-NAME) result in prevention of Cd stress induced oxidant damage in Arabidopsis and yellow lupine [16-17], but application of a NO donor sodium nitroprusside (SNP) decreases ROS accumulation in Cd stressed Brassica juncca and rice seedlings [18-19].”Page 2, second paragraph
Point 8: Page 13
3.5. nNOS transgenic plants showed increased melatonin levels under cadmium stress
Many studies have reported that melatonin plays a similar role to NO in tolerance of plants to Cd stress [32]. However, the relationship between NO and melatonin in response to Cd stress in plants is still unclear [33].
Results and conclusions from previously published reports should be included in Introduction.
Response: Thanks. We added one paragraph about the background of melatonin in Introduction in the revised paper as follow:
“Moreover, NO can also regulate hormone homeostasis such as indole-3-acetic acid (IAA) or melatonin (N-acetyl-5-methoxytryptamine), so as to alleviate Cd toxicity in plants [22-23]. Over past decades, increasing studies indicate that Cd stress can induce melatonin accumulation and exogenous melatonin improve Cd tolerance in different plants [24]. Melatonin is synthesized via four continual enzymatic reactions from tryptophan, requiring at least six enzymes: tryptophan decarboxylase (TDC), tryptophan hydroxylase (TPH), tryptamine 5-hydroxylase (T5H), N-acetylserotonin methyltransferase (ASMT), and serotonin N-acetyltransferase (SNAT)[24]. Many results showed that exogenous melatonin can alleviate Cd-induced oxidative damage by activating antioxidant systems [25-26], and decrease Cd accumulation by regulating the transcription of iron-transport genes [25,27]. Although melatonin and NO have similar role in response to Cd stress in plants, the relationship between them is still unclear.” Page 2
Point 9: Page 13
3.5. nNOS transgenic plants showed increased melatonin levels under cadmium stress
The relative expression levels of the genes of the melatonin metabolism process, including OsASMT1, OsTDC3, OsTDC1, and OsSNAT2, were increased under cadmium treatment in the transgenic lines compared with the WT lines (Figure 6C–F).
What do the abbreviations OsASMT1, OsTDC3, OsTDC1, and OsSNAT2 mean? What are the functions of the proteins encoded by these genes? This information should be included in Introduction.
Response: Thanks for the suggestion. We have included OsASMT1, OsTDC3, OsTDC1, and OsSNAT2 in revised Introduction as follow: “Melatonin is synthesized via four continual enzymatic reactions from tryptophan, requiring at least six enzymes: tryptophan decarboxylase (TDC), tryptophan hydroxylase (TPH), tryptamine 5-hydroxylase (T5H), N-acetylserotonin methyltransferase (ASMT), and serotonin N-acetyltransferase (SNAT)[24].” Page 2
Discussion
Point 10:Page 15
Recently, Many studies have reported the function of NO in response to Cd stress in plants. However, the role of NO in plant responses to Cd stress are still debated [6].
For example, Exogenous melatonin improved Cd tolerance by reducing NO accumulation resulting in lower Cd accumulation in Brassica pekinensis (Lour.)
Capital letter instead of lowercase.
Response: sorry for the mistake. We have changed “Many” to “many”, and “Exogenous” to “exogenous” in revised version.

Round 2
Reviewer 1 Report
Manuscript ID: agronomy-2458248 Revised version Accepted Thanks Manuscript ID: agronomy-2458248 Revised version Accepted Thanks